# Omadacycline: A Newly Approved Antibacterial from the Class of Tetracyclines

**DOI:** 10.3390/ph12020063

**Published:** 2019-04-21

**Authors:** Fernando Durães, Emília Sousa

**Affiliations:** 1Laboratory of Organic and Pharmaceutical Chemistry, Department of Chemical Sciences, Faculty of Pharmacy, University of Porto, Rua Jorge Viterbo Ferreira, 228, 4050-313 Porto, Portugal; fduraes5@gmail.com; 2CIIMAR, Interdisciplinary Centre of Marine and Environmental Research, University of Porto, Terminal de Cruzeiros do Porto de Leixões, Avenida General Norton de Matos P, 4450-208 Matosinhos, Portugal

**Keywords:** Omadacycline, aminomethylcycline, tetracycline, antibacterial, protein synthesis inhibitor, FDA approved

## Abstract

Omadacycline (Nuzyra®) is a new aminomethylcycline, approved by the U. S. Food and Drug Administration in 2018, as a tetracycline antibacterial. It can be used in community-acquired pneumonia and in acute bacterial skin and skin-structure infections. It was developed and is commercialized by Paratek Pharmaceuticals. It is a semisynthetic compound, derived from minocycline, capable of evading widely distributed efflux and target protection antibacterial resistance mechanisms and has demonstrated activity in a broad spectrum of bacteria.

## 1. Introduction

Antimicrobials are drugs capable of killing or inhibiting the growth of microorganisms. Their discovery, in the early 1900s, revolutionized medicine and public health, and largely contributed to an improvement in society’s quality of life [1]. However, the excessive and unruled use of antimicrobials, combined with bacteria’s genetic plasticity and rapid capacity of adaptation, led to the large-scale selection, emergence, and dissemination of antimicrobial resistance, which drastically reduced the efficacy of most antimicrobial drugs. Resistance mechanisms can be manifested through target modification, changes in the metabolic pathway, decrease in the drug uptake, and activation of efflux pumps [1,2]. Therefore, the search for new antimicrobials is an urgent issue and should be regarded as a priority.

In 2018, the Food and Drug Administration (FDA) approved 59 new drugs, of which 14 were anti-infectious agents. Within these, seven drugs were antiviral, two antiparasitic, four antibacterial and one a vaccine (Figure 1). From the four antibacterial drugs approved, Aemcolo® (Aries Pharmaceuticals) is a rifamycin, Zemdri® (Achaogen) is an aminoglycoside, and Nuzyra® (Paratek Pharmaceuticals) and Xerava® (Tetraphase Pharmaceuticals) are tetracyclines [3].

The aim of this review is to comprise the specific information regarding one of the tetracyclines approved by the FDA in 2018, Nuzyra® (omadacycline).

Omadacycline is an aminomethylcycline, a semisynthetic compound derived from tetracyclines. This family of drugs has been part of the antimicrobial therapeutics repertoire for over 60 years. Omadacycline, similarly to older tetracyclines, displays activity against a broad spectrum of bacteria, including Gram-positive, Gram-negative, anaerobic, and atypical pathogens. The main advantage over older tetracyclines resides in the fact that omadacycline is active against bacteria carrying the major efflux and target protection resistance determinants. These include not only strains resistant to tetracyclines, but also resistant to other antibiotics, such as quinolones, macrolides, and aminoglycosides [4].

## 2. Omadacycline

### 2.1. Names and Structure

Omadacycline (Figure 2, **1**), also known as PTK-0796 or (4S,4aS,5aR,12aR)-4,7-bis(dimethyl- amino)-9-[(2,2-dimethylpropylamino)methyl]-1,10,11,12a-tetrahydroxy-3,12-dioxo-4a,5,5a,6-tetrahy-dro-4H-tetracene-2-carboxamide, is the active principle of Nuzyra®, commercialized by Paratek Pharmaceuticals.

### 2.2. Uses

Omadacycline was approved by the FDA in October 2018, and it is prescribed for community-acquired bacterial pneumonia [5,6,7] and for acute bacterial skin and soft-tissue infections [8,9,10]. It is taken once daily, an advantage over minocycline and doxycycline, orally or intravenously, as a tosylate salt [5,8,11,12].

### 2.3. Synthesis

Omadacycline (**1**) was synthesized as part of a work developed by Honeyman et al. (2015) [13], with the aim of identifying novel minocycline (Scheme 1, **2**) derivatives at the C-9 position. The first step was the production of 9-aminomethylminocycline, through amidomethylation and deprotection of the resulting phthalimide, using methylamine. This resulted in a compound with a reactive, free amine, prone to molecular modifications. Studies were made in order to explore the differences in activity against resistant bacteria, using simple alkylamines, amides, carbamates, and ureas, among others. The synthetic pathway that led to the synthesis of omadacycline is schematized in Scheme 1 [13,14].

### 2.4. Mechanism Action and Resistance

Similarly to tetracyclines, omadacycline (1) acts through the inhibition of protein synthesis. It binds to 70S ribosomes with greater affinity than tetracycline. The modification in the C-9 position comes from the fact that tigecycline, a C-9 substituted semisynthetic derivative of minocycline, is not affected by the most common resistance mechanisms, namely efflux and ribosomal protection, while retaining activity. Concerning tetracycline efflux, studies with tigecycline implied that, although it induces the expression of efflux pumps, tigecycline is not a substrate for efflux transporters. Despite not being confirmed, the same was hypothesized for omadacycline. Ribosomal protection is not totally understood, but it is believed that ribosomal protection proteins can alter ribosome conformation, causing the release of tetracyclines. Taking this into account, there are two mechanisms by which omadacycline can evade ribosomal protection: An increased affinity for ribosome binding or omadacycline may bind in such a unique way that can circumvent ribosomal protection, as it has been suggested for tigecycline. This last possibility is supported by the fact that omadacycline binds with similar affinity as minocycline, which is susceptible to tet(M)-mediated ribosomal protection, in contrast to omadacycline. Furthermore, omadacycline is a C-9 modified tetracycline, which might make it possible for omadacycline to have additional binding sites, thus overcoming ribosomal protection while maintaining activity against tetracycline-resistant bacteria, as it happens with tigecycline [15].

Another mechanism of resistance of tetracyclines is mediated by tetracycline destructases, flavin-containing monooxidases capable of modifying tetracyclines through a variety of covalent modifications in their scaffold. The modification in omadacycline, which is similar to the modification in tigecycline, would probably also not lead to resistance against the tetracycline destructase enzymes, which is supported by the fact that omadacycline and tigecycline share similar activity profiles against other resistance mechanisms. Omadacycline also displayed a reduction of associated side effects, such as nausea, and an increase in antimicrobial activity. It has the same binding sites as tetracycline and tigecycline and is, therefore, susceptible to the same 16S rRNA mutations that confer binding site alterations. However, the two mutations that can alter the primary binding site in helix 34 or in the loop of helix 31, leading to tetracycline resistance impair the organism’s cell growth and only cause a low-level resistance [15,16,17,18,19]. 

### 2.5. Antibacterial Activity

Omadacycline has been tested in vitro and in vivo, and has demonstrated activity against Gram-positive and Gram-negative bacteria, as well as anaerobic bacteria. In terms of Gram-positive bacteria, omadacycline is active against *S. aureus* and methicillin-resistant *S. aureus*. It is also active against penicillin-resistant and multidrug-resistant *Streptococcus pneumoniae*, as well as vancomycin- and multidrug-resistant *Enterococcus*, and β-hemolytic *streptococci*. Omadacycline was shown to remarkably retain activity against isolates that were resistant to tetracycline [20,21].

Comparative studies were carried out by Macone et al. (2014) [4] against clinically significant bacteria, including bacteria containing known tetracycline resistance genes conferring ribosomal protection, tet(M), tet(O) and tet(S), and efflux, tet(K), and tet(L). Omadacycline’s activity was not only compared to that of tetracycline, doxycycline, and minocycline, but also to cephalosporins (cefotaxime), aminoglycosides (vancomycin), quinolones (levofloxacin), linezolid, macrolides (azithromycin), and lyncosamides (clindamycin) [4,22]. For *S. aureus*, omadacycline displayed better results than the other studied tetracyclines, such as tetracycline and doxycycline, implying that omadacycline is capable of circumventing tetracycline resistance in *S. aureus*. In fact, their minimal inhibitory concentration (MIC) range was 0.125–1 µg/mL for the tet(M) gene containing *S. aureus* and 0.125–0.25 µg/mL for the tet(K) containing strain, much lower than that of tetracycline, 32–>64 µg/mL and 16–32 µg/mL, respectively, and of doxycycline, with MIC ranges of 2–16 µg/mL and 1–4 µg/mL, respectively. It also displayed the best MIC ranges among the different classes of antibacterials, as well as MIC_50_ and MIC_90_. Omadacycline was also tested in methicillin-resistant *S. aureus* (MRSA), as well as in multidrug and methicillin-resistant *S. aureus*. When compared to all the antibacterials tested, omadacycline displayed better activity than most of the other antibacterials, doxycycline being the antibacterial with the best MIC_50_ for MRSA. For the multidrug and methicillin-resistant *S. aureus*, omadacycline was responsible for the best MIC ranges (0.25–0.5 µg/mL), MIC_50_ (0.5 µg/mL) and MIC_90_ (0.5 µg/mL) [4,22].

In the case of *Enterococcus faecalis* and *Enterococcus faecium*, which pose therapeutic challenges due to their acquired resistance mechanisms to vancomycin, omadacycline was also able to display good activity in both vancomycin resistant and susceptible *E. faecalis* and *E. faecium*, as well as in tetracycline resistant and susceptible strains. For the latter, different strains of *E. faecalis* and *E. faecium* were tested: Susceptible, ribosome protected (tet(M) and tet(S)), increased efflux (tet(L)) and both (tet(M) and tet(L)). Remarkably, omadacycline achieved a MIC of 0.25 µg/mL for the strain exhibiting both types of resistance, against a MIC of 16 µg/mL for doxycycline and >64 µg/mL for tetracycline. When only one type of resistance was expressed, omadacycline also displayed the best results: A MIC range of 0.125–0.5 µg/mL for tet(M) and 0.25–0.5 µg/mL for tet(S), against 32–64 µg/mL and 32 µg/mL, respectively, for tetracycline and 4–8 µg/mL and 2 µg/mL, respectively, for doxycycline; for tet(L), omadacycline displayed a MIC of 0.25 µg/mL, against 64 µg/mL for tetracycline and 16 µg/mL for doxycycline. In susceptible and multidrug-resistant strains of *E. faecalis*, omadacycline was also the best antibacterial, among different classes of antibacterials [4,22]. *E. faecium* was tested in the same way as the previous microorganism, and the results achieved were similar. For the ribosome protected strains, tet(M) and tet(O), omadacycline displayed a MIC range of 0.125–0.5 µg/mL and 0.12 µg/mL, respectively. For tetracycline, the results were, respectively, 32–64 µg/mL and 32 µg/mL, and for doxycycline, respectively, 2–8 µg/mL and 4 µg/mL. When it comes to the increased efflux strain, tet(K), omadacycline was able to present a MIC of 0.12 µg/mL, better than tetracycline’s 32 µg/mL and doxycycline’s 4 µg/mL. A double resistant strain was also tested, with both the tet(M) and tet(L) resistance genes expressed, and omadacycline presented a MIC of 0.25 µg/mL, being therefore more effective than doxycycline, with a MIC of 8–16 µg/mL, and tetracycline, with a MIC of >64 µg/mL. Omadacycline was also compared with antibacterials from different therapeutic classes in susceptible, vancomycin-resistant and multidrug- and vancomycin-resistant strains of *E. faecium*, and was by far the most effective in all the strains, displaying altogether the best MIC_50_ and MIC_90_ results [4,22].

*Streptococcus pneumoniae* is a respiratory pathogen growing concern due to its resistance to penicillins, cephalosporins, macrolides, and tetracyclines. Omadacycline displayed remarkable results against all the resistant strains, with MIC values of ≤0.06 µg/mL, for the ribosome protected strain expressing the tet(M) resistance gene, the penicillin-resistant strain, and tetracycline-, penicillin-, and azithromycin-resistant strains [4,22]. *S. pyogenes* and *S. agalactiae*, β-hemolytic *streptococci*, are also susceptible to omadacycline. Resistant strains were used, and omadacycline was able to inhibit their growth, with a MIC range of ≤0.06–0.5 µg/mL for the ribosome protected strain and ≤0.06–0.25 µg/mL for the increased efflux strain. When tested against other antibacterials, omadacycline had better results than minocycline and linezolid, but antimicrobials such as cefotaxime and the macrolides azithromycin and clindamycin displayed better MIC_50_ (≤0.06 µg/mL) than omadacycline (0.125 µg/mL) [4,22].

Regarding Gram-negative bacteria, omadacycline (**1**) was active against *Haemophilus influenzae*, *Klebsiella pneumoniae* and *Escherichia coli*. Studies were performed within these three species, in order to access the activity of omadacycline in comparison with other antibacterials. In this group, omadacycline displayed activity in vitro, but antibacterials such as cefotaxime, levofloxacin, and ciprofloxacin had better MIC_50_ results (≤0.06 µg/mL) than omadacycline (1 µg/mL for *E. coli* and *H. influenzae* and 2 µg/mL for *K. pneumoniae*). [4] For *E. coli*, the concentration of omadacycline needed to inhibit the growth of *E. coli* carrying the efflux gene tet(A) was also tested in comparison to other tetracyclines. Here, it was shown that omadacycline could inhibit *E. coli* strains with increased efflux at a concentration of 2 µg/mL, oppositely to tetracycline, which displayed a MIC range of 64–>64 µg/mL, and doxycycline, with a MIC of 16 µg/mL [4,20]. Other Gram-negative bacteria in which omadacycline has displayed activity are *Moraxella catarrhalis* (MIC_90_: 0.25 µg/mL), *Enterobacter aerogenes* (MIC_50_: ≤4 µg/mL), *Enterobacter cloacae* (MIC_50_: 2 µg/mL), *Serratia marcescens* (MIC_50_: 4 µg/mL), *Salmonella* spp. (MIC_50_: 2 µg/mL), *Shigella* spp. (MIC_50_: 1 µg/mL) and *Stenotrophomonas maltophilia* (MIC_50_: 2 µg/mL). *Legionella pneumophila* and *Chlamydia* spp., atypical bacteria, were also susceptible to omadacycline [20,22,23]. All serogroups of *L. pneumophila* were susceptible to omadacycline, with a MIC range of 0.06–1 µg/mL. Concerning *Chlamydia pneumoniae*, it has been shown that omadacycline is capable of inhibiting growth with a MIC range of 0.03–0.5 µg/mL, and displays the best MIC_50_ on par with azithromycin, which is 0.06 µg/mL [24]. Omadacycline also managed to inhibit the growth of *Acinetobacter baumanii*, with a MIC_50_ of 4 µg/mL and a MIC_90_ of 8 µg/mL [25].

Additionally, it has shown activity against anaerobes. In the scope of this study, omadacycline’s activity was compared to that of tigecycline, meropenem, moxifloxacin, clindamycin, metronidazole, and piperacillin-tazobactam. *Clostridium difficile*, *Clostridium perfringens* and *Peptostreptococcus* spp. are Gram-positive anaerobes, proven to be susceptible to omadacycline. For *C. difficile*, omadacycline displayed a MIC range of 0.25 to 8 µg/mL which is comparable to tigecycline’s (0.25 to 4 µg/mL). For these bacteria, both drugs display the same MIC_50_ of 0.25 µg/mL, but tigecycline displayed a more favorable MIC_90_, 0.25 µg/mL, as opposed to omadacycline’s 0.5 µg/mL. For *C. perfringens*, omadacycline displayed better activity than tigecycline, but less than meropenem or metronidazole. For *Peptostreptococcus* spp., omadacycline showed a MIC range of 0.06 to 2 µg/mL, and had the best results for MIC_50_, along with tigecycline, of 0.12 µg/mL, and MIC_90_, of 1 µg/mL [4,20,26]. In terms of Gram-negative anaerobes, omadacycline displayed activity towards some species of *Bacteroides* spp., *Prevotella* spp., and *Porphyromonas asaccharolytica*. Towards *Bacteroides* spp., the MIC ranged from 0.25 to 16 µg/mL for *B. fragilis*, 0.12 to 16 µg/mL for *B. thetaiotaomicron*, 0.06 to 2 µg/mL for *B. vulgatus* and 0.06 to >16 µg/mL for *B. ovatus*; towards *Prevotella* spp., omadacycline displayed a MIC range of 0.12–8 µg/mL (not as much as meropenem, with a MIC range of 0.03–1 µg/mL); towards *P. asaccharolytica*, omadacycline displayed a MIC range of 0.06 to 2 µg/mL, and had the same MIC_50_ and MIC_90_ as tigecycline, of 0.25 µg/mL and 0.5 µg/mL, respectively [20,26].

*Mycoplasma* spp. and *Ureoplasma* spp. are pathogens capable of causing community-acquired pneumonia and urogenital conditions in children and immunosuppressed adults. These microorganisms have been progressively associated with antimicrobial resistance, and resistance to macrolides, fluoroquinolones, and tetracyclines has been reported [27,28]. Omadacycline was compared to tetracycline, doxycycline, azithromycin, and moxifloxacin. Against *M. hominis*, omadacycline presented the lowest MIC_50_ and MIC_90_ values, 0.032 µg/mL and 0.063 µg/mL, respectively. Additionally, its activity was unaffected by the presence of the tet(M) gene. Concerning *M. pneumoniae*, omadacycline displayed a MIC_90_ of 0.25 µg/mL, being surpassed by moxifloxacin (0.125 µg/mL). In strains highly resistant to macrolides, omadacycline displayed remarkably low MIC values, of 0.125 µg/mL, in contrast to azithromycin’s 16 µg/mL. For *Ureoplasma* spp., omadacycline was slightly less active (MIC_50_ of 1 µg/mL and MIC_90_ of 2 µg/mL). However, these MICs remained consistent in macrolide-, tetracycline- and moxifloxacin-resistant strains, ranging from 0.25 to 2 µg/mL, even though doxycycline displayed lower MIC values in tetracycline-resistant *Ureoplasma* [29].

Studies have been performed in order to compare omadacycline against tigecycline, concerning human respiratory tract pathogens. Their MIC_90_ was calculated, and omadacycline had comparable activity to tigecycline, displaying the same results in methicillin-susceptible *S. aureus* (0.25 µg/mL) and *M. pneumoniae* (0.12 µg/mL), and better results for *L. pneumophila*, with a MIC_90_ of 0.25 µg/mL, against tigecycline’s 8 µg/mL. Although tigecycline displayed slightly better results in *S. pneumoniae*, MRSA, *M. catarrhalis*, *H. influenzae* and *C. pneumoniae*, it also led to the manifestation of a higher number of individuals with side effects, such as headache, nausea and vomiting [30,31].

Omadacycline has also shown promising activity against fast-growing mycobacteria. Rapidly growing mycobacteria cause infections preferentially in immunosuppressed individuals. *Mycobacterium abscessus*, *M. chelonae* and *M. fortuitum* pose therapeutic challenges, as they are not easily treated with antimicrobials used for other mycobacterial infection. In this scope, omadacycline was compared to tigecycline and doxycycline, and it was demonstrated that omadacycline and tigecycline had similar activity against *M. abscessus*, with a MIC_50_ of 1 µg/mL and a MIC_90_ of 2 µg/mL. These results are much better than those of doxycycline, whose MIC values were >64 µg/mL. In the case of *M. chelonae*, omadacycline displayed a MIC_50_ of 0.125 µg/mL, while tigecycline has a lower value of 0.06 µg/mL. However, both drugs had the same MIC_90_, which was 0.25 µg/mL. Finally, for *M. fortuitum*, omadacycline displayed the lowest MIC_50_ of 0.125 µg/mL, against tigecycline’s 0.25 µg/mL. Once again, both drugs had the same MIC_90_, 0.5 µg/mL. In the three cases, omadacycline and tigecycline presented much lower MIC values than doxycycline and amikacin, which were used as control drugs [32].

### 2.6. Clinical Studies

Phase I studies have been carried out in order to access the pharmacokinetics and dosage of omadacycline. A wide range of doses have been investigated, both orally and intravenously. This study showed the good tolerability of omadacycline in both ways of administration. It was shown that doses higher than 300 mg, in the case of intravenous administration, would lead to an increase, albeit reversible, of alanine aminotransferase. In the case of oral administration, doses higher than 400 mg led to mild nausea [20].

For acute skin and skin structure infections, phase II and phase III trials were carried out. Phase II studies were aimed at the comparison between omadacycline and linezolid, with or without aztreonam. These randomized, controlled, investigator-blind, multicenter studies included patients that had one of four complicated skin and skin structure infections: Wound infection, major abscess, cellulitis, or an infected ulcer in the lower extremity. Both drugs started being administered intravenously, with a dosage of 100 mg daily of omadacycline vs 600 mg twice a day of linezolid, with the option of transitioning to an oral dosage of 200 mg daily of omadacycline versus 600 mg twice a day of linezolid. Conclusions drawn from these studies included the comparable results between omadacycline and linezolid in terms of safety and tolerability, highlighting the higher rate of success in patients treated with omadacycline rather than with linezolid, and also for *Staphylococcus aureus*. After these phase II studies, omadacycline was claimed to be non-inferior to linezolid [33].

Consequently, omadacycline advanced for phase III studies. The “Omadacycline in Acute Skin and Skin Infections Study” (OASIS-1), a double-blind, randomized, multicenter study had the aim of accessing the safety and efficacy of omadacycline in a once-daily, intravenous dose, with the option of transitioning to oral administration, against a twice-daily dosage of linezolid. The patients started to receive 100 mg of intravenous omadacycline twice a day, followed by 100 mg of intravenous omadacycline once a day, or 600 mg of intravenous linezolid twice a day, with the option to switch to oral omadacycline (300 mg once daily) or linezolid (600 mg twice a day), for seven to fourteen days. Efficacy results were consistent across trial groups, lesion sizes, and causative pathogens, which included MRSA and *E. faecalis*. This study also proved the non-inferiority of omadacycline in comparison to linezolid, demonstrating both drugs to have similar safety and side-effect profiles [5,8,34]. OASIS-2 was a randomized study aimed at the comparison of oral omadacycline versus linezolid, with a duration of seven to fourteen days. For the omadacycline group, patients first received 450 mg daily for the first two days, and 300 mg/day in the following days. Apart from meeting the primary and end points for the study, omadacycline was also successful in maintaining high rates of clinical success for all Gram-positive microorganisms and overperformed linezolid in *S. aureus* [21,34].

Regarding community acquired bacterial pneumonia, phase III studies were performed. The “Omadacycline for Pneumonia Treatment in the Community” (OPTIC) study was a randomized, double-blind, multicenter study, aimed at the evaluation of omadacycline’s safety and efficacy against moxifloxacin for the treatment of adults with community-acquired bacterial pneumonia. In this study, patients with clinical evidence of community-acquired bacterial pneumonia, radiographically confirmed acute bacterial pneumonia, and signs of infection or systemic inflammatory response were included. Omadacycline was administered intravenously, with a dosage of 100 mg twice a day for two doses, followed by 100 mg daily, intravenously, against a dosage of 400 mg once a day of moxifloxacin, intravenously, for three days, with the option to switch to oral administration. In terms of pathogens, omadacycline displayed a similar profile as moxifloxacin for atypical pathogens, Gram-negative bacteria and *Streptococcus pneumoniae*. Even though omadacycline had slightly lower success rates with *S. aureus*, it was still considered non-inferior to moxifloxacin [5,21].

Currently, there are two additional clinical studies registered in ClinicalTrials.gov regarding omadacycline: One study aimed at the comparison between oral omadacycline and nitrofurantoin for the treatment of cystitis [35], and another with the objective of accessing intravenous or oral omadacycline against levofloxacin for the treatment of acute pyelonephritis [36]. Preliminary phase I studies of omadacycline in women with cystitis have already been performed. Within this randomized, open-label study, patients have received different doses of omadacycline, both orally and intravenously. This had the purpose of evaluating the pharmacokinetics of omadacycline, as well as their microbiological response. The main difference between this study and previous studies resides in the fact that adverse effects in the gastrointestinal system were higher than expected, such as vomiting and nausea, even though they were mild and temporary, and did not lead to withdrawals from the trial. However, further studies in larger groups are required for more assertive conclusions to be drawn [37].

## 3. Conclusions

Omadacycline, a new aminomethylcycline from the tetracycline family, has displayed interesting antibacterial results. This newly approved drug has shown potent activity against a broad-spectrum of infectious bacteria, including Gram-positive, Gram-negative, anaerobes and atypical, even if multidrug resistance is present. Another great advantage is the fact that omadacycline can be taken orally and once daily. So far, it has been indicated for bacterial skin and skin-structure infections and community-acquired bacterial pneumonia, but its indication can be extended to other infections, since clinical trials on urinary tract infections are ongoing. Its safety and efficacy has been proven through clinical trials, as well as omadacycline’s non-inferiority to currently used drugs.

Taking this into account, and adding the fact that omadacycline can circumvent common tetracycline resistance mechanisms, such as efflux and ribosome protection, it definitely constitutes an exciting addition to the antimicrobial armamentarium.

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
