# Peer review of "Omadacycline: A Newly Approved Antibacterial from the Class of Tetracyclines"

_pharmaceuticals, 2019, doi:10.3390/ph12020063_

Round 1
Reviewer 1 Report
Major points
The manuscript is classified as a review. In my opinion, it is rather a sketch or profile of omadacycline, than a review or even a mini-/micro-review. Clearly, more information should be added here. The literature in PubMed yields 57 citations for the search term “omadacycline”, roughly 3 times the number of publications cited here.
Antibacterial activity, lines 106-117: This section must be expanded. What is clearly lacking, is an analysis of the activity of omadacycline in comparison to tetracycline, minocycline, doxycycline and tigecycline for susceptible and resistant strains. This is the most important and also most interesting part of the manuscript, because it directly addresses the useability and potential applicability of omadacycline. There is much detailed information in the literature that should be used here. It should also be mentioned that omadacycline has shown promising activity against fast-growing mycobacteria (Shoen et al. Antimicrob Agents Chemother 2019 doi: 10.1128/AAC.02522-18). More focus should be placed on what makes omadacycline an interesting and exciting addition to the antibiotic portfolio.
Minor points
Abstract, line 17: I would include here that omadacycline evades “widely distributed efflux and target protection” antibacterial resistance mechanisms, because it does not protect against the rRNA mutations and, most likely, in analogy to tigecycline won’t protect against the tetracycline destructases.
Introduction, line 26: “allied” implies a willful association of two partners. Since antimicrobial misuse and bacterial genetic plasticity are unrelated and independent, I would rather use a word like “combined” or “”added to”.
Introduction, line 27: Antimicrobial resistance is much older than therapeutic antibiotic use. This has been convincingly shown several times. I would therefore not use the word “development” here, but rather something like “led to the large-scale selection, emergence and dissemination of antimicrobial resistance”.
Mechanism of action, line 61: please change to “is not affected by the most common resistance mechanisms, …”.
Mechanism of action, lines 65-67: What is meant by “two mutations are necessary”? In E. coli, the combination of the tetracycline-resistant mutations in helix 34 and in the loop of helix 31 have not been combined yet, despite numerous efforts. In both helix 34 and in the helix 31 loop, only a single mutation is necessary to confer a tetracycline-resistant phenotype. Please clarify or correct.
Mechanism of action, line 67: Even though it is speculative, the modification in omadacycline, which is similar to the modification in tigecycline, would probably also not lead to resistance against the tetracycline destructase enzymes. This should be mentioned.
Antibacterial activity, line 111: change “tetracyclin” to “tetracycline”.
Author Response
REVIEWER 1
Major points
The manuscript is classified as a review. In my opinion, it is rather a sketch or profile of omadacycline, than a review or even a mini-/micro-review. Clearly, more information should be added here. The literature in PubMed yields 57 citations for the search term “omadacycline”, roughly 3 times the number of publications cited here.
We understand the reviewer concern and would like to change the submission to a Brief Report instead of a Review type of Manuscript. Nevertheless we added more information concerning omadacycline activity and trials.
Antibacterial activity, lines 106-117: This section must be expanded. What is clearly lacking, is an analysis of the activity of omadacycline in comparison to tetracycline, minocycline, doxycycline and tigecycline for susceptible and resistant strains. This is the most important and also most interesting part of the manuscript, because it directly addresses the useability and potential applicability of omadacycline. There is much detailed information in the literature that should be used here. It should also be mentioned that omadacycline has shown promising activity against fast-growing mycobacteria (Shoen et al. Antimicrob Agents Chemother 2019 doi: 10.1128/AAC.02522-18). More focus should be placed on what makes omadacycline an interesting and exciting addition to the antibiotic portfolio.
We thank the reviewer for these comments. We added both information of resistance and activity profiles and hope with these modification to have highlighted the added value of omadacycline.
Minor points
Abstract, line 17: I would include here that omadacycline evades “widely distributed efflux and target protection” antibacterial resistance mechanisms, because it does not protect against the rRNA mutations and, most likely, in analogy to tigecycline won’t protect against the tetracycline destructases.
We changed the manuscript accordingly.
Introduction, line 26: “allied” implies a willful association of two partners. Since antimicrobial misuse and bacterial genetic plasticity are unrelated and independent, I would rather use a word like “combined” or “”added to”.
We changed the manuscript accordingly.
Introduction, line 27: Antimicrobial resistance is much older than therapeutic antibiotic use. This has been convincingly shown several times. I would therefore not use the word “development” here, but rather something like “led to the large-scale selection, emergence and dissemination of antimicrobial resistance”.
We changed the manuscript accordingly.
Mechanism of action, line 61: please change to “is not affected by the most common resistance mechanisms, …”.
We changed the manuscript accordingly.
Mechanism of action, lines 65-67: What is meant by “two mutations are necessary”? In E. coli, the combination of the tetracycline-resistant mutations in helix 34 and in the loop of helix 31 have not been combined yet, despite numerous efforts. In both helix 34 and in the helix 31 loop, only a single mutation is necessary to confer a tetracycline-resistant phenotype. Please clarify or correct.
We thank the reviewer for identifying this lapse and changed the manuscript accordingly.
Mechanism of action, line 67: Even though it is speculative, the modification in omadacycline, which is similar to the modification in tigecycline, would probably also not lead to resistance against the tetracycline destructase enzymes. This should be mentioned.
We thank the reviewer the suggestion and added this hypothesis in the revised manuscript.
Antibacterial activity, line 111: change “tetracyclin” to “tetracycline”.
We thank the reviewer for identifying this lapse and changed the manuscript accordingly.
Reviewer 2 Report
The review manuscript by F. Durães and E. Sousa reviewed the development of a newly approved antibiotics named omadacycline. It is excited to see that omadacycline displayed very broad-spectrum antibacterial activity, aganist both Gram-positive and Gram-negative bacteria as well as anaerobic bacteria. However, the current review did not describe the background very well. Other documental summary on the mechanism, the clinical tries, and the antibacterial activities should be reviewed in good details. A conclusion in the end is strongly recommended. In line 72, “phtalimide” should be “phthalimide”.
In general, the reviewer dose not think the review was well written, much detailed review need to be done. The reviewer suggests acceptance of the manuscript only if with major revision.
Author Response
The review manuscript by F. Durães and E. Sousa reviewed the development of a newly approved antibiotics named omadacycline. It is excited to see that omadacycline displayed very broad-spectrum antibacterial activity, aganist both Gram-positive and Gram-negative bacteria as well as anaerobic bacteria. However, the current review did not describe the background very well. Other documental summary on the mechanism, the clinical tries, and the antibacterial activities should be reviewed in good details.
We thank the reviewer for these comments. We added both information of resistance and activity profiles and hope with these modification to have highlighted the added value of omadacycline.
A conclusion in the end is strongly recommended. In line 72, “phtalimide” should be “phthalimide”.
We thank the reviewer for identifying this lapse and changed the manuscript accordingly.
In general, the reviewer dose not think the review was well written, much detailed review need to be done. The reviewer suggests acceptance of the manuscript only if with major revision.
We understand the reviewer concern and would like to change the submission to a Brief Report instead of a Review type of Manuscript. Nevertheless we added more information concerning omadacycline activity and trials.
Round 2
Reviewer 1 Report
The manuscript is much improved in its revised form. There are several points (mostly editorial) that still have to be taken care of.
Major comments:
lines 41-44: The main destinction between omadacycline and older tetracyclines is that omadacycline is active against bacteria carrying the major efflux and target protection resistance determinants. Both omadacycline and the older tetracyclines are active against a broad spectrum of bacteria. Please modify.
lines 95-96: replace the sentence after "by the fact that omadacycline" with and tigecycline share similar activity profiles against the other resistance mechanisms.
The observation that omadacycline overcomes tetracycline resistance does not forcibly imply that it will act by the same mechanism. This assumption is made rather by similarities between tigecycline and omadacycline in both their chemical structures and their resistance profiles against efflux, ribosome protection and ribosome modification determinants.
Minor comments:
line 38: change omadocycline to omadacycline.
line 40: insert "repertoire" between "therapeutics" and "for".
line 76: replace "substiuted" with "substituted".
line 87: replace "oppositely" with "in contrast".
line 91: change the beginning of the sentence to: "Another mechanism of resistance to tetracyclines is mediated by tetracycline destructases, ..."
line 100: replace "organism" with "organism's".
line 109: replace "retained" with "retain".
line 125: change to "doxycycline being".
line 131: change "For this latte" to "For the latter"
line 132: change "ribosomal protected" to either "ribosome protected" or "ribosomally protected".
line 153: replace "that has gathered" woth "of growing".
line 159: see comment to line 132.
lines 160-161: Minocycline is not another antibacterial class than omadycycline. Replace mybe "antibacterials from other classes" with "other antibacterials".
line 166: replace "scope" with "group".
line 171: replace "In this scope" with "Here".
line 178: replace "serogroup" with "serogroups".
line 179: The genus designation Chlamydophila is not really used anymore. Please swith back to "Chlamydia".
line 200: replace "tigegycline" with "tigecycline".
line 210: replace "In highly resistant to macrolides strains" with "In strains highly resistant to macrolides, "
line 214: delete "the".
line 238: replace "has" with "have".
line 246: replace either "These" with "this" or "study" with "studies".
line 256: replace the 1st sentence with "Consequently, omadacycline advanced to phase III studies.
line 266: delete "The" before OASIS-2.
line 270: replace "microorganism" with "microorganisms".
lines 282, 283: write Streptococcus pneumoniae and S. aureus in italics. Please also switch back to a non-bold typeface.
line 298: replace "showed" with "shown".
Author Response
We thank the reviewer for the valuable comments and modify the revised manuscript accordingly.
The manuscript is much improved in its revised form. There are several points (mostly editorial) that still have to be taken care of.
Major comments:
lines 41-44: The main destinction between omadacycline and older tetracyclines is that omadacycline is active against bacteria carrying the major efflux and target protection resistance determinants. Both omadacycline and the older tetracyclines are active against a broad spectrum of bacteria. Please modify.
lines 95-96: replace the sentence after "by the fact that omadacycline" with and tigecycline share similar activity profiles against the other resistance mechanisms.
The observation that omadacycline overcomes tetracycline resistance does not forcibly imply that it will act by the same mechanism. This assumption is made rather by similarities between tigecycline and omadacycline in both their chemical structures and their resistance profiles against efflux, ribosome protection and ribosome modification determinants.
We thank the reviewer for these comments and modify accordingly in the revised manuscript.
Minor comments: ALL THE FOLLOWING COMMENTS WERE REVISED/CHECKED.
line 38: change omadocycline to omadacycline.
line 40: insert "repertoire" between "therapeutics" and "for".
line 76: replace "substiuted" with "substituted".
line 87: replace "oppositely" with "in contrast".
line 91: change the beginning of the sentence to: "Another mechanism of resistance to tetracyclines is mediated by tetracycline destructases, ..."
line 100: replace "organism" with "organism's".
line 109: replace "retained" with "retain".
line 125: change to "doxycycline being".
line 131: change "For this latte" to "For the latter"
line 132: change "ribosomal protected" to either "ribosome protected" or "ribosomally protected".
line 153: replace "that has gathered" woth "of growing".
line 159: see comment to line 132.
lines 160-161: Minocycline is not another antibacterial class than omadycycline. Replace mybe "antibacterials from other classes" with "other antibacterials".
line 166: replace "scope" with "group".
line 171: replace "In this scope" with "Here".
line 178: replace "serogroup" with "serogroups".
line 179: The genus designation Chlamydophila is not really used anymore. Please swith back to "Chlamydia".
line 200: replace "tigegycline" with "tigecycline".
line 210: replace "In highly resistant to macrolides strains" with "In strains highly resistant to macrolides, "
line 214: delete "the".
line 238: replace "has" with "have".
line 246: replace either "These" with "this" or "study" with "studies".
line 256: replace the 1st sentence with "Consequently, omadacycline advanced to phase III studies.
line 266: delete "The" before OASIS-2.
line 270: replace "microorganism" with "microorganisms".
lines 282, 283: write Streptococcus pneumoniae and S. aureus in italics. Please also switch back to a non-bold typeface.
line 298: replace "showed" with "shown".